# From a Pinecone to Design of an Active Textile

**DOI:** 10.3390/biomimetics5040052

**Published:** 2020-10-13

**Authors:** Veronika Kapsali, Julian Vincent

**Affiliations:** 1London College of Fashion, University of the Arts London, London WC1V 7EY, UK; 2Nature Inspired Manufacturing Centre, School of Engineering, Heriot-Watt University, Edinburgh EH14 4AS, UK; jv21@hw.ac.uk

**Keywords:** hygronastic movement, design research, botany, smart textiles, active materials

## Abstract

Botanical nastic systems demonstrate non-directional structural responses to stimuli such as pressure, light, chemicals or temperature; hygronasty refers to systems that respond specifically to moisture. Many seed dispersal mechanisms such as wheat awns, legume pods, spruce and pinecones fall within this classification. The variety of behaviours varies greatly from opening and closing to self-digging, but the mechanism is based on differential hygroscopic swelling between two adjacent areas of tissue. We describe the application of hygronastic principles specifically within the framework of textiles via the lens of structural hierarchy. Two novel prototypes are presented. One is designed to increase its permeability to airflow in damp conditions and reduce permeability in the dry by 25–30%, a counterintuitive property compared to conventional cotton, wool and rayon textiles that decrease their permeability to airflow as their moisture content increases. The second prototype describes the design and development of a hygroscopic shape changing fibre capable of reducing its length in damp conditions by 40% when compared with dry.

## 1. Introduction

The functionality of clothing is complex, multifaceted and the object of significant scholarly activity. Flugel [1] was among the first scholars to distinguish between the psychological and physiological aspects of clothing. This research is concerned with physiological aspects, specifically the role of garments as a barrier between the external environment and the microclimate of the space between the skin and the inner face of a garment system. When overheated, we sweat. Glands in the skin secrete a dilute salt solution onto the skin which cools the body surface by evaporation. Overheating can be caused by a high external temperature or metabolic heat caused by physical activity.

The key factors that affect the air and heat flow properties of clothing are the materials they are made of (e.g., textiles, embellishments, components), the design features of the garment (e.g., the location of openings, pockets, overall fit, etc.) and the combination of garments being worn. In current practice, the wearer manages the properties of the clothing system behaviourally, by adding or removing layers or opening or closing fastenings to maintain physiological comfort. Such comfort is controlled by both objective (permeability of the clothing to heat, moisture and air) and subjective (size, fit and aesthetic aspects of the garment such as drape) factors [2]. Here we focus on objective factors.

Most of the effort over the last thirty years has been in monitoring body functions and adapting the clothing accordingly. For instance, next to skin garments such as T-shirts and baselayers feature components capable of monitoring and analyzing biological functions. International lifestyle brand Ralf Lauren brought to market the “Polo Tech” shirt in 2015 [3]. SME Flexwarm incorporate flexible, washable heating systems into scarves and jackets [4] that are controlled via an application on mobile phones (Figure 1). Such devices are controlled by the user/wearer and require external energy delivered via a battery to power textile heating elements or minature components capable of sensing heart rate. The ideal situation might be a garment system that can adapt its properties autonomously, without intervention from the wearer. Such a smart textile would be a cloth-based system capable of sensing force, temperature, moisture, pH or magnetic field that would trigger or inform an actuation which could be the delivery of a signal, capturing of a data point, physical, mechanical or chemical response.

Non-electronic smart textile components, in the form of microcapsules or fibres (Figure 2), produced by Outlast Technologies [5], rely on phase change materials sensitive to temperature changes either in the microclimate or the external environment. The materials release or absorb heat as the phase changes between liquid and solid.

Hollies [6] studied descriptors of discomfort and noted repeating descriptors such as clammy, damp and clingy among numerous independent studies. These terms have since been used by the industry to describe the sensation caused by the concentration of moisture in the microclimate of clothing. Our skin does not feature a distinct mechanism for moisture perception.

A study seeking to understand the relationship between the presence of microclimate moisture vapour, temperature and sensation of dampness, by the wearer in an urban context [7], concluded that a decrease in skin temperature (caused by the evaporation of sweat) is not exclusively linked to damp discomfort sensations. According to Cabanac [8], the effect or “pleasantness” of a thermal stimulus is determined by one’s internal state. Therefore, if an individual is hot, a cold stimulus will be perceived as pleasant and vice versa. However, moisture vapour in the microclimate is directly linked to the sensation of damp, an observation in line with several studies conducted in controlled environments. Thus, microclimate moisture presents a more effective signal of damp discomfort than temperature.

This paper explores a practice-based approach to understanding how a hygroscopic shape change in nature can inform the design of a novel textile system to promote physiological comfort. The criterion is that the textile autonomously adapts its resistance to air flow in response to the moisture content of the microclimate between the skin and the textile, increasing its permeability in response to the production of perspiration, thus facilitating its evaporation and cooling. Thus the sensor and the effector in this system are one and the same—the yarn in the textile.

## 2. The Pinecone Effect

Most plants are made from individual cells, each of which has a stiff cell wall that can resist a tensile force. By controlling the concentration of osmolar effectors (mostly hexoses) within the cell cytoplasm, the cell absorbs water and so generates an internal pressure, acting against the cell wall. This generates the stiffness and mechanical stability of the plant. The cell wall may also be impregnated with lignin which stabilises the cellulose fibres and reduces the requirement for internal pressure for control of their shape. By controlling the amount and distribution of water the plant can control shape and provide the force that drives changes in size and shape. Thus, plants demonstrate a wide variety of motion from subtle nutation to violent eruption, all without muscles. This is generally referred to as nastic movement and is responsible for numerous functions essential to survival such as seed dispersal and energy conservation [9,10,11,12,13,14].

The structure of plants and the control of their movement presents significant opportunities for biomimetics, especially in the field of active material systems and soft robotics because it provides an entirely novel view on what constitutes a machine and how these can be powered. Nastic movement is achieved entirely by drawing on alternative yet abundant sources of energy, such as light (photonasty/nyctinasty), temperature (thermonasty) and touch (seismonasty) to trigger physical responses. We focus on hygronasty—plant movement triggered by the manipulation of moisture.

### 2.1. Hygronastic Behaviour in Seed Pods

Plants are not able to migrate from one location to another but have evolved strategies that ensure their seeds or spores are disseminated in specific conditions that optimize their chances for germination. Anemochorous seeds such as those in pinecones and the seed pods of legumes and geraniums rely on wind for their dispersal. In dry conditions the hygroscopic structures protecting mature seeds open up, sometimes explosively expelling the seeds with force, or enabling the seeds to be carried away by the wind. In damp conditions the structures close and prevent the seeds becoming damp before they are ready to germinate. Comprehensive studies of this phenomenon have been carried out on pinecones [15,16,17,18], legume valves [19,20] geranium pods [7] and seed hairs [21,22,23] and reveal that all these mechanisms are based on imbibition; the hygroscopic contraction and swelling of plant tissues as they lose or imbibe water, respectively.

The actuation of hygronastic behaviour is due to the structure of the cell walls of the moving part—opposing forces created by differences in hygroscopic swelling between cells or cell parts [7,19] similar to the function of a bimetallic strip when exposed to changes in temperature [15]. In the pinecone bract there are two types of sclerenchyma cell (Figure 3): one tissue type is long thin strands of fibres packed together, the other is a cellular arrangement composed of short rectangular cells with thick cell walls known as sclerids. A sorption analysis of the two tissue types revealed no significant difference in the amount of water absorbed from a humid atmosphere, yet the swelling between the two differed greatly. Dawson, Vincent and Rocca [15] found that the sclerids have a coefficient of hygroscopic expansion three times greater than the long thin sclerenchyma cells. Reyssat and Mahadevan [18] found a 20% difference in length between damp and dry sclerid cells when compared using an environmental scanning electron microscope.

An examination of the cellulose orientation within the two types of tissue using polarised light microscopy showed that the cellulose microfibrils in the walls of the long sclerenchyma cells are orientated at approximately 30° to the main cell axis while the orientation of cellulose microfibrils in the sclerids is 74–90° [15] to the main axis. This difference in orientation can account for the difference in the amount of longitudinal swelling. Wardrop and Allen [17] using X-ray diffraction found that the lattice spacing of the crystalline regions was the same in both wet and dry samples. However, the microfibrils were wider apart in the amorphous regions of the wet samples showing that the shrinking and swelling of the cell walls can be attributed to the desorption and sorption of moisture from the matrix separating the cellulose microfibrils.

Microfibril orientation is a significant and general factor in hygronastic behaviour. For instance, it controls the dehiscence (splitting) of the two halves (valves) of the legume pod. At maturity and in dry conditions, the valves split along a seam to expose the seeds. Depending on the legume species, the valves split either by bending outwards or twisting around their individual axes. This reversible motion (the valves return to their original form when exposed to moisture) is managed by the orientation of cellulose microfibrils within the valve cell walls. This configuration is abstracted as two separate adjoining layers, such as a bimetallic strip. Bending occurs when the cellulose microfibrils are orientated at right angles to each other, while twisting is caused by two possible configurations—first, the microfibril orientation is at 45° to the valve axis and the others are orientated in parallel, and the second consists of fibrils at 45° to the valve axis and those in the adjoining layer are positioned in reflection symmetry [19]. Figure 4 provides a conceptual model summarizing the relationship between cellulose microfibril orientation and observed hygronastic movement.

### 2.2. Pinecone Effect Textiles

The application of this mechanism into a textile prototype was demonstrated by Dawson, Vincent and Rocca [15]. They identified a commercial textile composed of a tightly woven polyester fabric on the face (the side of the fabric which is intended to be used as the surface or visible in the end product) lined by a non-porous hydrophilic polyether-ester block copolymer membrane such as those marketed under the brand Sympatex^TM^ (Figure 5). The two materials are connected using an industrial lamination process using hydrophilic polymeric adhesives. Such laminates are used in garment systems to allow moisture vapour to pass out through the textile but prevent external water droplets penetrating the clothing system. Polyester fibres swell slightly when exposed to moisture; the polyether component swells significantly in the presence of moisture. “U”-shaped die cuts, positioned across the plane of the bilayer fabric, open (Figure 5b) and close (Figure 5a) as moisture increases and decreases. Thus, the layer becomes porous.

On the basis of this published work, Nike (sportswear brand) independently developed this bilayer principle as a double-jersey textile structure; a knitted fabric made using a machine that features a double set of needles to produce two parallel layers of fabric joined by interlocking stitches. The active textile system was implemented in terms of yarn fibre composition. The face (top, outward facing layer) of the textile used yarns containing hydrophobic polyester fibres. The second layer, or back of the textile, was composed of yarns containing fibres made from hydrophilic polyamide. The polyamide fibres expand when damp [24]. The resulting system was implemented in a tennis dress. Scales, similar to the U-shaped flaps in the pinecone effect textile, were introduced at the back of the dress; the resulting technology was marketed under the trade name Macro React (Figure 6).

## 3. Structural Hierarchy of Textiles

Most practitioners in constructed textiles work at the level of the yarn, while printers and finishers work with completed textile structures. They rarely work with the structural hierarchy of the textile, considering how a mix of properties in the yarn can affect the function of the textile. This presents a key obstacle in implementing biomimetic concepts in the design of textiles, since nearly all biological materials are composites that derive their properties from structural interactions within their hierarchy.

The implementation of the pinecone effect described above requires a part of the textile to move out of the plane of the textile, requiring the textile to have free space above its surface. In a tennis dress the textile is the only layer in use, which is also true of light summer clothing such as T-shirts. However, in everyday clothing the adaptive behaviour can easily be obstructed by a covering layer or by accessories such as handbags, scarves, belts, etc. Therefore, the wearer will be unable to personalise their ”look” or carry anything other than in pockets incorporated into the clothing system. Application in the under-arm or crotch areas, the warmest areas of the body, is not suitable since there is little free surface area.

Having demonstrated that a textile can be created that increases its permeability with moisture, the need is to create a yarn that becomes thinner or shorter in damp conditions, leaving space between the individual strands. This could operate within the plane of the textile and not be inhibited by design features and ergonomics of structures or layers above or below the plane of the textile. Such a textile would increase its permeability to gases as it became damp.

### 3.1. Hygronastic Yarn: Approach to Design and Production

A yarn is typically made from fibres and/or filament(s) with or without a twist, the key distinguishing feature being the high length to width ratio. The simplest form of yarn is an untwisted, unsupported slit filament. Lurex Company Ltd., Leicester, UK makes yarns from films 12–40 μm thick that are slit into fine filaments with a width varying from 0.25 to 0.8 mm. These continuous strips of film can be used directly as monofilament yarns and are referred to as unsupported yarns [25]; however, these yarns typically are not strong and so have limited application. Tensile strength is improved by twisting the monofilament with another filament yarn such as polyester; these yarns are referred to as supported. The unsupported monofilament yarn structure demonstrated the simplest approach to transform a bicomponent film, with engineered hygronastic behaviour, into a yarn, for direct application into a constructed textile.

A prototype bilayer film was created by Kapsali [7] using two different forms of cellulose. A polyvinyl alcohol (Ghosenolgh 20/ GH20, Ashland Global Speciality Chemicals, Delaware, US), was used for the swelling component and an ethylcellulose (Hercules Aqualon r EC-N10 0100, Ashland Global Speciality Chemicals, Delaware, US), for the non-swelling part. A 25 m length of film was produced on a roll-to-roll pilot facility at MacDermid Autotype, Oxford, UK, slit into 0.8 mm-wide monofilaments and wound onto bobbins for processing into textiles by Lurex Company Ltd., Leicester, UK.

Owing to their low tensile strength, unsupported monofilament yarns are not used alone in the warp (lengthwise threads in a woven textile) but introduced mainly in the weft (threads applied widthwise, at 90° to the warp, of a woven textile). A novel woven structure was designed to host the adaptive monofilament yarn and minimise restrictions on the potential shape change of the monofilament introduced at interlacing points between warp and weft yarns. A standard spun polyester yarn was used in the warp and the active monofilament in the weft. The textile prototype (Figure 7) was produced on a handloom, with technical support from the woven textile department at Chelsea College of Art and Design, part of the University of Arts London.

### 3.2. Approach to Dynamic Air Permeability Testing

Standard air permeability tests on clothing textiles are conducted in controlled environments using a set temperature and relative humidity (RH); the testing procedure is defined in BS 5636. The absorption of moisture by the fibre in a textile changes the dimensions of the fibre leading to changes in the structural parameters of the fabric such as thickness and porosity. This is most noticeable in the more hygroscopic fabrics and is observed in both woven and non-woven fabrics (Figure 8). Key variables include the fabric structure and intrafabric constraint factors, such as the number of bonding points, yarn twist, and yarn cross-over points [26].

Gibson et al. [27] developed a specialised test rig to measure the effects of atmospheric moisture levels on the airflow resistance of textiles. The principles used for measuring the sorption behaviour of materials (changes in sample weight plotted against relative humidity) were applied to the design of the rig and testing protocol, where airflow resistance is plotted against relative humidity. The test rig comprises a cell with a controllable temperature and relative humidity, a circular clamp for holding a small textile sample (approximately 5 cm in diameter), a pump generating a known pressure, and pressure sensors at the face and back of the sample. With a textile sample clamped into the cell, the temperature and humidity are set and the sample is allowed to equilibrate. Once the sample reaches equilibrium, gas is pumped through the fabric and the pressure drop across the fabric is measured, hence the resistance of the fabric to gas flow. These measurements are taken at a set temperature and varied humidity to provide a profile of how the textile alters its resistance to gas flow at different levels of humidity.

The sample was clamped onto a specialised cell, part of the test rig. The test sequence was set using a computer interface. Set points of 4 h were programmed for each relative humidity (default setting of 0.0, 0.2, 0.4, 0.6, 0.8, 0.9., and 0.95 relative humidity) and the temperature was set at 30 °C. The gas flow rate was set at 2000 mL/min and the diameter of the sample was 2.5 cm giving an area of 4.9 cm^2^. Given the sensitive nature of the active film, the highest relative humidity was 95% to avoid damaging the sample.

### 3.3. Results and Discussion

At above 50–60% RH, the flat monofilament yarn rolls up along its axis to form a cylinder 0.5 mm in diameter, leaving space between adjacent filaments. This increase in spacing should allow the resistance to air flow to be less. This behaviour is novel. All conventional hygroscopic textiles such as those made from wool, cotton, silk and other natural fibres increase resistance to airflow as humidity increases, while those made from synthetics such as polyamide and polyester remain relatively unaffected.

The results (Figure 9) show this reduction in resistance to air flow by approximately 20% at 80% RH. The the correlation coefficient r = −0.86 indicates a strong downhill linear relationship between the decrease in airflow resistance and increase in relative humidity. In practice this means that in higher humidities the textile sample is more permeable to air than in dry conditions. Although, this constitutes sufficient proof of concept for the prototype sample; the processing and materials used are not suitable for industrial processing. The creation of a commercial grade yarn that demonstrates hygronastic functionality requires the development of a novel fibre.

For comparison with standard commercial textiles, two woven textile samples, one made with cotton, the other with Nylon 66, were tested using the same protocol. The scale, density and overall design of these prototypes were engineered to frame the functionality of the active yarn and were not comparable with a commercial textile.

The cotton fabric (Figure 10a) gradually increased in resistance until 60% RH; above 60% RH, the resistance increased quite sharply. This is expected and in line with known properties of cotton.

The nylon fabric (Figure 10b) slightly increased in airflow resistance until 40% RH, then resistance decreased until 80% RH was reached, where it reached the original air flow resistance of the dry fabric. Polyamide fibres, such as 6 and 6.6, increase in length as they absorb moisture, causing minor changes to the sample’s dimensions which can cause a tightly clamped sample to sag and interfere with the accuracy of gas flow measurements.

The cotton and nylon samples were tightly woven textiles with a high number of yarn cross-over points per cm^2^, which makes the airflow resistance of both samples relatively high. The cotton fabric would be the most comfortable to wear as its hygroscopicity functions wicks moisture away so that the moisture in the microclimate next to the wearer’s skin will be low for a time depending on the wearer’s activity and sweating. Soon after the wearer begins to perspire, the swelling fibres will reduce the gaps between the yarns which results in an increase in air flow resistance. The fabric then becomes a barrier trapping saturated air within the microclimate that diffuses into the saturated areas of the clothing. This is a highly undesired effect and is directly related to damp, clammy sensations.

Although the nylon sample demonstrates less fluctuation in airflow resistance during exposure to higher levels of environmental moisture, the fact that it is not as hygroscopic as natural fibres means that as moisture in the microclimate builds up, the tightly woven structure will retain saturated air beneath the clothing and cause discomfort. The industry has developed various approaches to mitigating discomfort from moisture build up. Tri- or multilobal fibre architectures (e.g., Dri-Fit brand) are introduced into the design of the cross section of synthetic fibres as wicking mechanisms that draw moisture from the back to the face of the textile, where it can evaporate.

## 4. Hygronastic Fibre

The next iteration of this mechanism focused on the development of a commercial grade hygronastic fibre. The hypothesis is that this will enable a wider scope for design within the broader context of a commercially viable textile system.

The application of the hygronastic model at the yarn-scale demonstrated the desired functionality; however, the chemistry and yarn morphology were deemed unsuitable for commercial application. Although the chemistry could have been developed into a combination suitable for garment use, the overall design and size of the monofilament yarn was not appropriate for application in industrial textile production systems. Although the pursuit of commercially viable chemistry for this specific iteration was of little value, the application of the active mechanism into the basic building block of commercial textiles, a fibre, informed the next iteration. An active fibre would enable the application of the hygronastic mechanism across a broader scope of textile structures as indicated in Figure 11. In fibre form, the active mechanism can be implemented into knitted and woven textiles via yarns and directly into non-woven textiles; however, the reverse is not possible—i.e., the implementation of the mechanism at textile scale into yarn or fibre.

Figure 12 illustrates the approach to the research and development strategy mapped out by MMT Textiles Ltd. (a private research and development company co-founded by Kapsali specifically to advance the adaptive textile technology within the commercial sector) who acquired the intellectual property for this research in 2009. The overall aim is to bring to market a range of commercially viable textiles with adaptive thermal and/or airflow resistance in response to changes in moisture vapour levels in the environment. In line with the bilayer model applied to the previous fabric and yarn prototypes, a bicomponent fibre architecture was identified, and a range of cross section archetypes (Figure 13) and polymer ratios were considered. Additionally, a range of commodity polymers used in fibre production was reviewed based on their hygroscopic behaviour, melt temperature, modulus of elasticity and compatibility with other polymers within a bicomponent fibre structure.

The eccentric sheath core and side by side configurations were identified as the most effective for shape change. The polymer review resulted in the selection of four potential materials, Aqualon (AQ55s) Ethylcellulose standard grade (Hercules Incorporated), Nylon 6 (N6), polypropylene (PP) and polyester (PET)—the two former polymers were selected for their hygroscopic expansion properties, and the latter for its resistance to hygroscopic expansion.

A summary of the polymer and archetype combinations used to create sample fibres for testing is presented in Table 1. Analysis of samples based on processability (how successful the polymer combinations could be processed using the bicomponent fibre extrusion machinery into viable products). The samples containing AQ55s were deemed unsuitable for commercial fibre spinning, and the resulting fibres became tacky when exposed to moisture. We did not try a N6 PET combination since these fibres become longer as they absorb moisture [28] and we were seeking a contraction in damp conditions (Figure 14). The remaining N6 PP combination processed well by the industrial fibre co-extrusion and provided an ideal starting point from a commercial perspective because both polymers can be sourced on the open market as commodity materials.

The next step was to identify the polymer ratio that delivered the most dimensional difference in length between damp and dry and whose results were consistent across numerous damp–dry cycles. Two denier fibre samples with 50/50, 60/40 and 70/30 N6/PP ratios were measured by conditioning samples in a controlled humidity environment (airtight jars containing saturated potassium sulphate solutions with 95% relative humidity at 25 °C, and silica beads that change colour when saturated with moisture vapour). The most successful combination was the 70/30 which demonstrated an average of 23% difference in length between damp and dry, while the 60/40 samples averaged 20% across repeated cycles of exposure to damp–dry conditions.

## 5. Conclusions

Biomimetic approaches to design can be relevant to textile practice. The same as Velcro, adjustments to conventional textile manufacture methods can result in the translation of a mechanism from biology into technology. However, this does require an “out of the box” approach and a willingness of highly skilled experts in various technical aspects of the textile industry’s value chain to review and challenge their tacit knowledge.

This work demonstrates that it is possible to apply learning from the hygronastic mechanisms observed in plant seed dispersal to the design and production of both commercial and experimental textile prototypes with active open and closing mechanism that responds to environmental moisture levels. What it does not prove is how much a textile needs to alter its airflow resistance to affect the physiological comfort or the market opportunity for such adaptive textiles. Instead, the example of the Nike tennis dress, in particular, demonstrates the importance of design and application of an active mechanism within a garment system as a key factor in the development of this technology.

The next step in the development of this specific approach to the further development of textiles that demonstrate the pinecone effect is to use the fibre in yarn prototypes that can be produced using conventional manufacturing processes. The active mechanism both in the original pinecone effect textile work of Dawson, Vincent and Rocca [15] and later Nike [24] is implemented at the fabric level of a textile’s structural hierarchy. The developments described in this paper demonstrate that the introduction of an active behaviour at the fibre level provides wider opportunities for design and implementation in textile products—the higher up the hierarchy the mechanism is implemented, the more restricted the application. The authors’ view is that efforts to implement the active mechanism lower down the hierarchy will allow a greater degree of flexibility and opportunity for research and experimentation within textile practice.

Overall, the work highlights that the textile industry presents an untapped beneficiary; there are significant opportunities for the implementation of mechanisms and structures using textile methods and processes. These are not limited to the introduction of novel properties and behaviours but potentially increase the sustainability and circularity of the way we design, manufacture and dispose of textile products. As such, the work described in this paper constitutes a case study evidencing the emergence of biomimetic textiles as an new area of study.

## 6. Patents

The following patents have resulted from the work reported in this paper. Granted: Canadaand 2716700, China—ZL200880127912.8, Japan—2010-548163, USA—US-2015-0140886-A1, Europe—2861790.

## Figures and Tables

**Figure 1 biomimetics-05-00052-f001:**
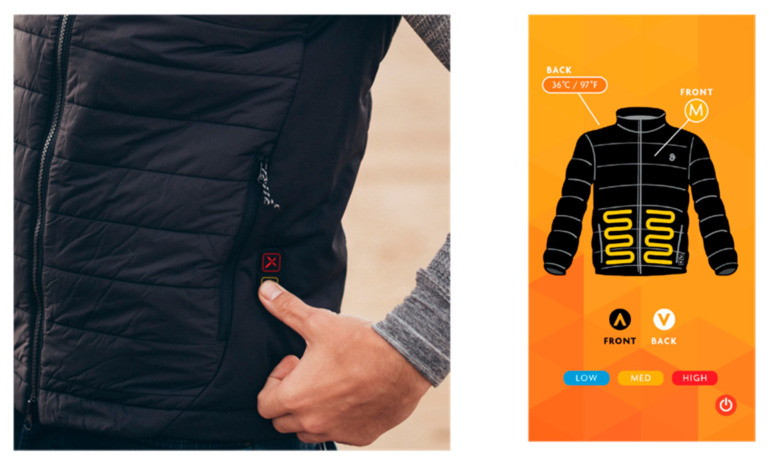
Flexwarm jacket detail and mobile phone application source: 8kFlexwarm.

**Figure 2 biomimetics-05-00052-f002:**
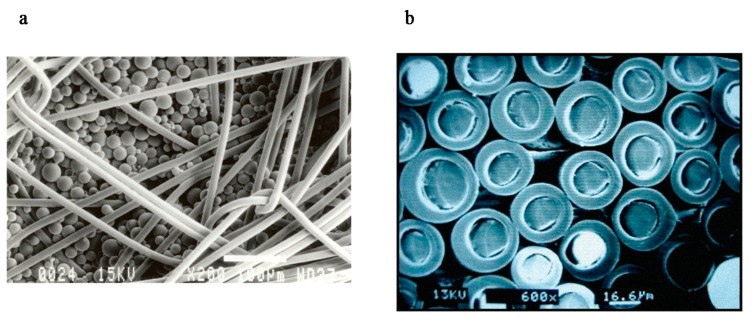
Phase change microcapsule applied onto textile as a coating (**a**), magnification 200×. Phase change material incorporated into the core of a fibre (**b**), magnification 600×. Source: Outlast Technologies.

**Figure 3 biomimetics-05-00052-f003:**
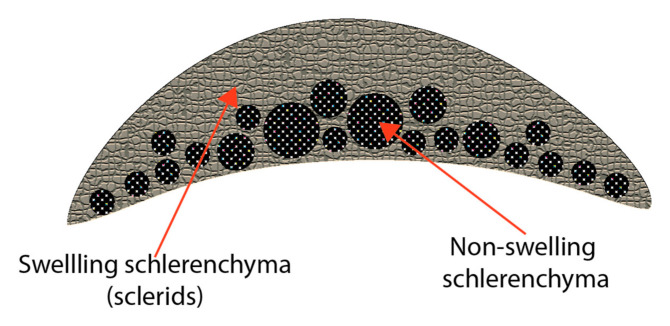
Stylised representation of pinecone bract cross section illustrating the location and distribution of the two types of schlerenchyma fibre.

**Figure 4 biomimetics-05-00052-f004:**
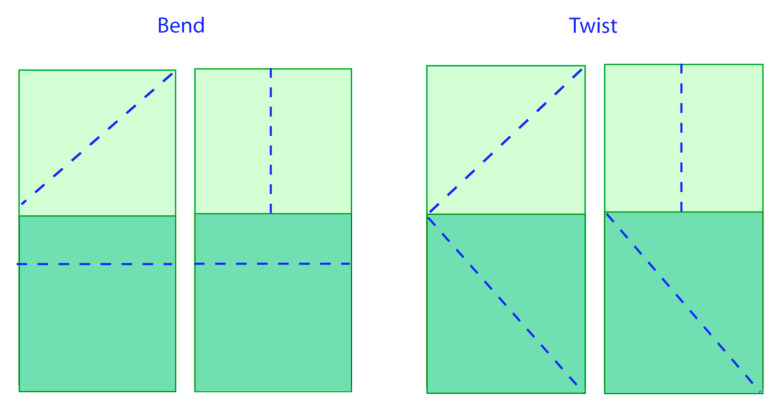
Abstracted model of the relationship observed between cellulose orientation and shape change in hygronastic seed pods source: Kapsali [7].

**Figure 5 biomimetics-05-00052-f005:**
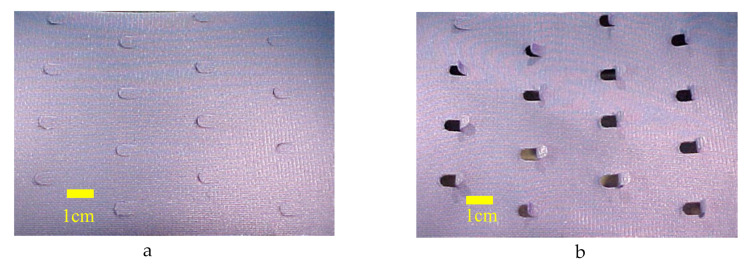
Photograph of original pinecone effect textile in dry (**a**) and damp (**b**) conditions. Sample size is 10 × 5 cm. Pores open when the hygroscopic polyether layer swells on exposure to moisture vapour.

**Figure 6 biomimetics-05-00052-f006:**
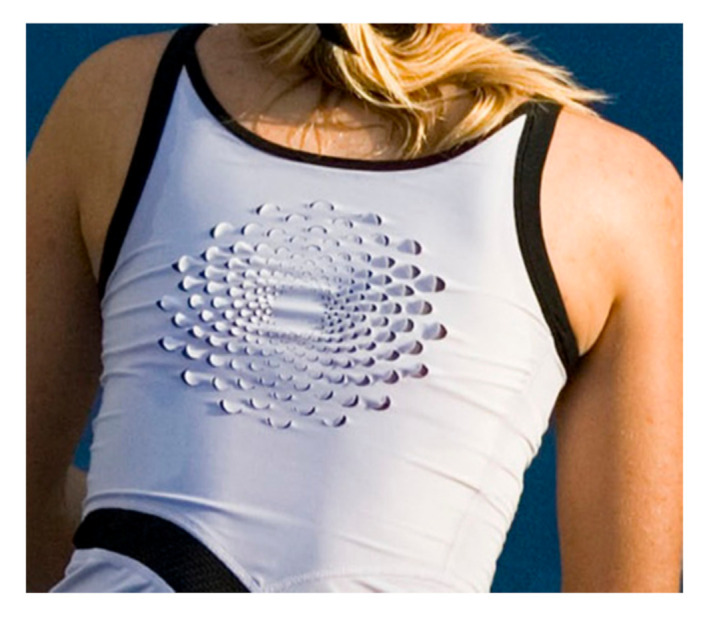
Detail of Nike Micro React sports wear garment active double cloth system in damp conditions. Source: Getty Images.

**Figure 7 biomimetics-05-00052-f007:**
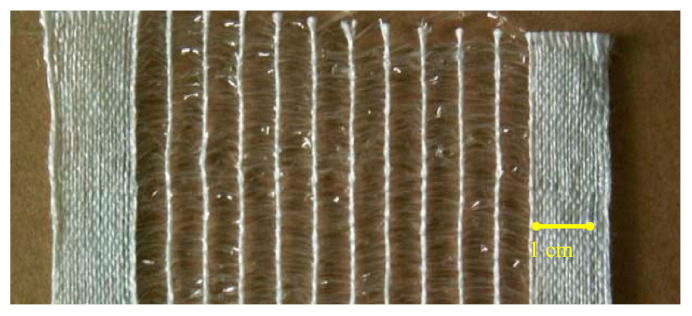
Photograph of dry active textile detail. The active bilayer yarn constitutes the clear strips in the weft (horizontal) of the fabric and the white thread is the polyester yarn of the fabric warp (vertical).

**Figure 8 biomimetics-05-00052-f008:**
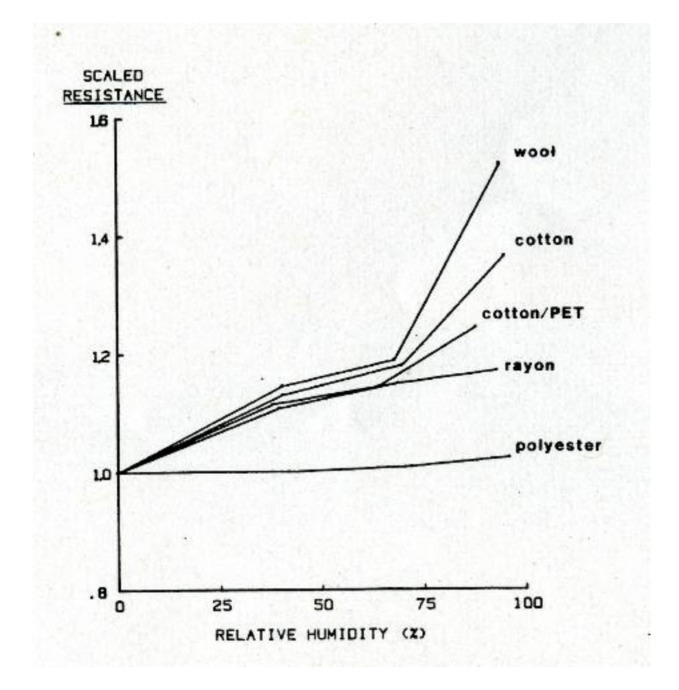
Effect of moisture absorption on air flow resistance through a variety of conventional fabrics [15].

**Figure 9 biomimetics-05-00052-f009:**
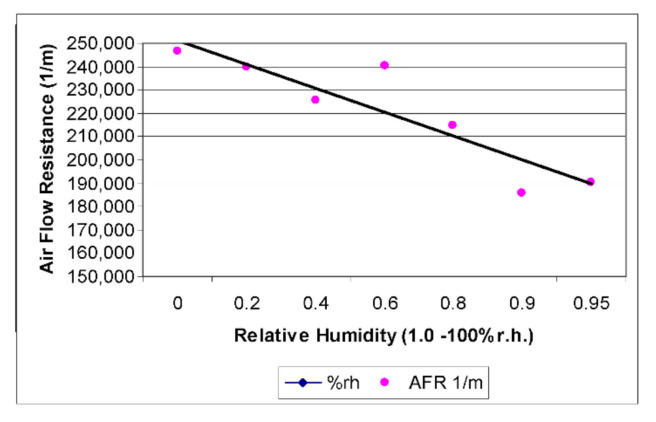
Active yarn textile prototype test results. Textile reduces resistance to airflow as humidity increases correlation coefficient r = −0.86.

**Figure 10 biomimetics-05-00052-f010:**
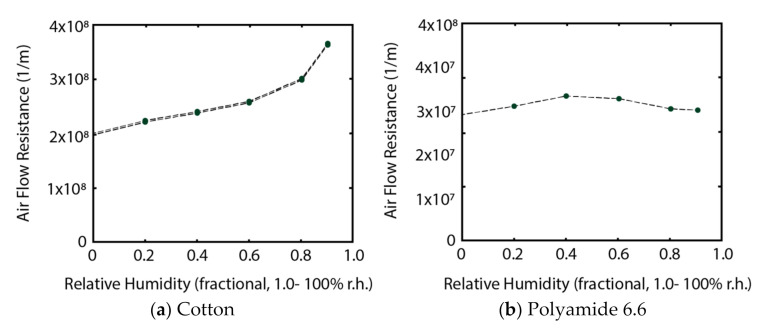
Results from cotton and nylon samples tested using by Phil Gibson, US Army Natick Soldier Systems, Natick. (**a**). 100% Cotton. Weight: 150 g/m^2^, source: Paolo Gilli, cira: Spring/Summer 2004 reference: Sirena, Width: 148–150 cm (**b**). 100% Polyamide 6.6 Weight: 83g/m^2^, source: William Reed, circa: 2001, reference: 131980, width: 155 cm.

**Figure 11 biomimetics-05-00052-f011:**
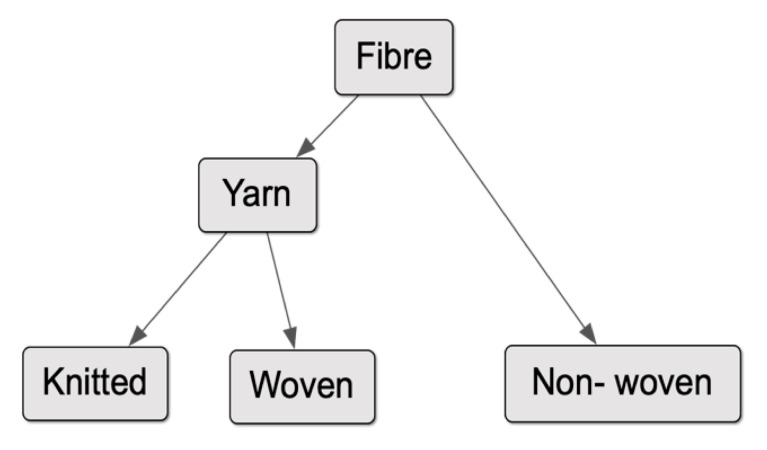
Structural map of basic textiles based on scale—fibres are the basic building block that can be applied to all fabric structures; i.e., knitted and woven textiles via yarns and directly into non-wovens.

**Figure 12 biomimetics-05-00052-f012:**
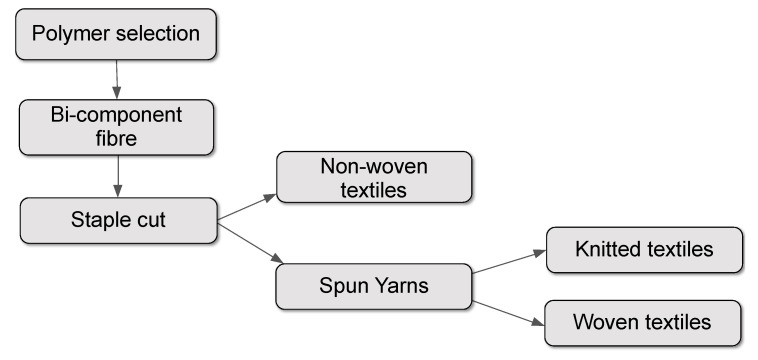
Process map describing research and development scope developed by MMT Textiles Ltd.

**Figure 13 biomimetics-05-00052-f013:**
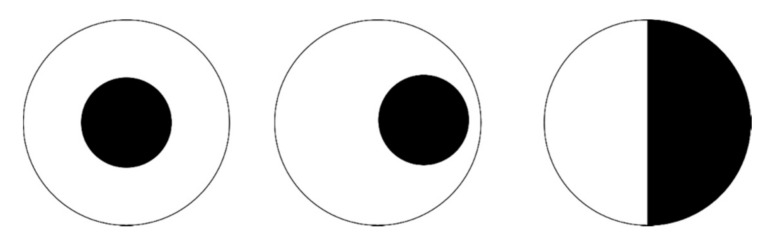
Examples of bicomponent fibre archetypes, from left to right: sheath core, eccentric sheath core and side by side.

**Figure 14 biomimetics-05-00052-f014:**
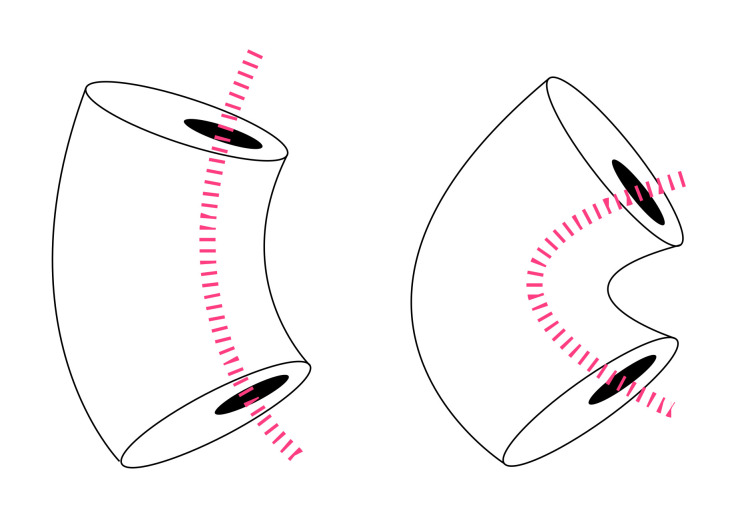
Visualisation of active fibre behaviour in dry (**left**) and damp (**right**) conditions. The antagonistic forces created between the Nylon 6 (N6), which elongates in damp conditions and the stiff, hydrophobic polypropylene (PP) cause the fibre to bend, defined as an increase in angel of crimp in the textile industry.

**Table 1 biomimetics-05-00052-t001:** Overview of cross section design and polymer combinations applied to the development of fibre samples. Key: AQ55s (Ethlylcellulose), N6 (Nylon 6), PET (Polyester), PP (Polypropylene), ESC (eccentric sheath core), SXS (side by side).

Ratio.	Composition	Cross Section
50/50	N6 PP	ESC
70/30	N6 PP	ESC
60/40	N6 PP	ESC
70/30	AQ55s PP	ESC
60/40	AQ55s PP	ESC
70/30	AQ55s PP	ESC
50/50	AQ55s PP	ESC
50/50	AQ55s PP	ESC
50/50	AQ55s PP	ESC
50/50	AQ55s PP	ESC
50/50	AQ55s PP	ESC
50/50	AQ55s PET	SXS
50/50	AQ55s PET	ESC
50/50	AQ55s PET	ESC
50/50	AQ55s PET	SXS
70/30	AQ55s PET	SXS
50/50	AQ55s PET	ESC
70/30	AQ55s PET	ESC
70/30	AQ55s PET	ESC

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
