# Peer review of "From a Pinecone to Design of an Active Textile"

_biomimetics, 2020, doi:10.3390/biomimetics5040052_

Round 1
Reviewer 1 Report
This is an interesting paper that clearly demonstrates a novel innovation. I have a few comments (detailed below).
L.38-49. Can the authors provide a reference (e.g. patent etc for the Ralph Lauren, Flexwarm and Outlast technologies?
L.157-158. Reference to patents etc?
L.196 Fig 3- does source need to be referenced?
L. 201-202 – as above – Lurex- any patents?
L. 306 How was the regression analysis performed? What is the significance level?
L. 347 Figure 7 is of poor resolution
Author Response
L.38-49. Can the authors provide a reference (e.g. patent etc for the Ralph Lauren, Flexwarm and Outlast technologies?
The text has been updated to provide more detail and images on the developments discussed
L.157-158. Reference to patents etc?
References to patents have been updated
L.196 Fig 3- does source need to be referenced?
Source is now referenced
L. 201-202 – as above – Lurex- any patents?
References to patents have been updated
L. 306 How was the regression analysis performed? What is the significance level?
This has been updated and explained in the text
L. 347 Figure 7 is of poor resolution
A new image of higher resolution has been used to replace this
Reviewer 2 Report
Thank you for an interesting and relevant article that focuses on a field that has not been much explored. Thus, the article can be considered as an important contribution to the further development of research within biomimetics as well as textile technology.
The comments are primarily related to the structure of the article and how core aspects and insights are disseminated.
- Introduction:
- No specific comments
- The Pinecone Effect
- Even though the mechanical effect of the pinecone is described in text, it would be very helpful with one or more illustrations. This would, at least to me, make it much easier to follow. This could e.g. support the sections ‘l70-l85’ and ‘l131-l143’.
- Would it be possible to add one or more references to the section ‘l70-l85’? It seems appropriate considering the nature of information provided.
- Should the footnote be a footnote or be in round bracket like in the rest of the article? Are these explanations necessary?
- I propose to Figure 1 to before or after the section in which it is described (either before l131 or after l142. It disturbs the reading flow as it is now.
- The same as above for Figure 3, e.g. after l165.
- I propose to introduce one or more sub-sections (2.1-2.x) to make the different aspects the section is looking into, more apparent.
- Structural hierarchy of textiles
- Figure 3: It looks like a) and b) have been rotated 90°
- The figure text to Figure 4 is very long. Most would make sense to integrate in the body text. Figure 4 could be moved to after the section that ends on l237.
- The section: 3.1. Hygronastic yarn is generally difficult to read. Introducing titles on section could help the reading flow:
- It would be good to introduce a title on the section l224-l237, e.g. ‘Test material’.
- A title for the sections l255-l261+l262-l273 could help clarify the content of these; this could e.g. be ‘Test method’.
- Furthermore, is it necessary to describe the test methods this detailed?
- Hygronatic fibre
- No specific comments
- Conclusion
- The conclusion could elaborate more on specific insights from the experiment(s) performed
Overall:
- Different reference systems seem to have been used
- It could help the overall ‘visual experience’ of the article to align the different visual means. Maybe this is already part of the publication procedure?
Author Response
- The Pinecone Effect
- Even though the mechanical effect of the pinecone is described in text, it would be very helpful with one or more illustrations. This would, at least to me, make it much easier to follow. This could e.g. support the sections ‘l70-l85’ and ‘l131-l143’.
- Would it be possible to add one or more references to the section ‘l70-l85’? It seems appropriate considering the nature of information provided.
- Should the footnote be a footnote or be in round bracket like in the rest of the article? Are these explanations necessary?
- I propose to Figure 1 to before or after the section in which it is described (either before l131 or after l142. It disturbs the reading flow as it is now.
- The same as above for Figure 3, e.g. after l165.
- I propose to introduce one or more sub-sections (2.1-2.x) to make the different aspects the section is looking into, more apparent.
I have introduced new images and subsections as advised by the reviewer
- Structural hierarchy of textiles
- Figure 3: It looks like a) and b) have been rotated 90°
- The figure text to Figure 4 is very long. Most would make sense to integrate in the body text. Figure 4 could be moved to after the section that ends on l237.
- The section: 3.1. Hygronastic yarn is generally difficult to read. Introducing titles on section could help the reading flow:
- It would be good to introduce a title on the section l224-l237, e.g. ‘Test material’.
- A title for the sections l255-l261+l262-l273 could help clarify the content of these; this could e.g. be ‘Test method’.
- Furthermore, is it necessary to describe the test methods this detailed?
I have introduced new images and subsections as advised by the reviewer
- Conclusion
- The conclusion could elaborate more on specific insights from the experiment(s) performed
I have included some additional thoughts and insights from the work.